# Electrical and Reliability Characteristics of Self-Forming Barrier for CuNd/SiOCH Films in Cu Interconnects

**Yi-Lung Cheng [1],\*, Chih-Yen Lee [1], Wei-Fan Peng [1], Giin-Shan Chen [2] and Jau-Shiung Fang [3]**

[1] Department of Electrical Engineering, National Chi-Nan University, Nan-Tou 54561, Taiwan; ck7766@gmail.com (C.-Y.L.); tommy860211@gmail.com (W.-F.P.)

[2] Department of Materials Science and Engineering, Feng Chia University, Taichung 40724, Taiwan; gschen@fcu.edu.tw

[3] Department of Materials Science and Engineering, National Formosa University, Huwei 63201, Taiwan; jsfang@nfu.edu.tw

\* Correspondence: yjcheng@ncnu.edu.tw; Tel.: +886-49-2910960-4987; Fax: +886-49-2917810

**Abstract:** In this study, Cu-2.2 at. % Nd alloy films using a co-sputtering deposition method were directly deposited onto porous low-dielectric-constant (low-*k*) films (SiOCH). The effects of CuNd alloy film on the electrical properties and reliability of porous low-*k* dielectric films were studied. The electrical characteristics and reliability of the porous low-*k* dielectric film with CuNd alloy film were enhanced by annealing at 425 °C. The formation of self-forming barrier at the CuNd/SiOCH interface was responsible for this improvement. Therefore, integration with CuNd and porous low-*k* dielectric is a promising process for advanced Cu interconnects.

**Keywords:** porous low-*k* dielectric; CuNd; self-forming barrier; Cu; reliability

## 1. Introduction

For back end of line interconnects in advanced integrated circuits (ICs), copper (Cu) and porous low-dielectric-constant (low-*k*) materials have been widely used to reduce the resistance–capacitance (RC) time delay and improve operating performance [1–3]. To avoid the direct contact of Cu and porous low-*k* materials, a barrier is required to prevent Cu diffusion into the dielectric film [4,5]. This barrier increases wiring resistance and the increasing magnitude grows as the IC critical dimension continues to shrink. This is caused by a significant increasing ratio of the cross-sectional area of a barrier to the total wire, as the thickness of a barrier remains unchanged to maintain an equivalent barrier capacity [6]. For advanced ICs, the increasing height-to-width (aspect) ratio for via and trench structures makes the conformal deposition of a barrier and no-void filling of Cu film more challenging [7].

To solve these issues for 32 nm technological nodes and beyond with a further smaller feature size, the chemical vapor deposition (CVD), or atomic layer deposition (ALD), method was proposed to replace the traditional physical vapor deposition (PVD) method [8,9]. These methods can provide better step coverage or sidewall integrity. However, the main issue is the penetration of the used deposition precursor via the open surface pores of the porous low-*k* material. The accompanying problems are high leakage currents and severely degraded reliability [10].

A self-forming barrier is a promising strategy for Cu barrier scaling because it only needs one Cu–*M* (*M* = Mn, Mg, or Al) alloy layer instead of two layers (Ta/TaN barrier and Cu seed layer). Firstly, a Cu–*M* alloy layer is directly deposited on the dielectric film. Next, a heat treatment is carried out to form a $MO_x$ barrier at the interface [11–13]. Most studies developed self-forming barriers on $SiO_2$ films [11–15]. However, few studies related to self-forming barriers on porous low-*k* dielectric films

were reported [16]. A different result is expected because its interface is different from SiO$_2$ film, demonstrating a need to study the electrical characteristics and reliability of the porous low-*k* dielectric film as it integrates with a self-forming barrier.

Therefore, this study provides a procedure for a self-forming barrier by depositing a CuNd alloy layer onto a porous low-*k* dielectric film and then annealing. The effects of the formed self-forming barrier on the electrical characteristics and reliability of porous low-*k* dielectric films are investigated.

## 2. Experiments

The porous low-*k* dielectric films used in this study were deposited on *p*-type (100) silicon substrates by a plasma-enhanced chemical vapor deposition method (applied material producer tool). Diethoxymethylsilane (DEMS) and oxygen (O$_2$), as the film's matrix precursors, and α-terpinene (ATRP), as an organic porogen precursor, were introduced into the reactor. During the deposition, the temperature, pressure, and power were 300 °C, $1.0 \times 10^4$ Pa, and 600 W, respectively. After deposition, ultraviolet (UV) thermal-assisted curing at 350 °C was performed for 300 s to remove the organic porogen to form the pores in the film [17]. This produced a porous low-*k* dielectric film with a SiOCH composition. The thickness of porous low-*k* SiOCH films was controlled at ~130 nm, which was determined using an optical-probe system with an ellipsometer (Film Tek$^{TM}$ 3000SE, High Point, NC, USA). The pore size and porosity of the porous low-*k* SiOCH film were 1.4 nm and 15.0%, respectively, which were determined from the isotherm of ethanol adsorption and desorption using ellipsometric porosimetry (Semilab, Mode PS-1100, Budapest, Hungary).

Next, metal-insulator-silicon (MIS) capacitors were fabricated by depositing a CuNd layer onto the surface of the porous low-*k* SiOCH film through a metal mask. A CuNd alloy film was deposited by a co-sputtering method using a CuNd target (RF power supply, Hsinchu, Taiwan) and a Cu target (DC power supply, Hsinchu, Taiwan). The purity of both targets was 99.9%. In the CuNd target, the Nd concentration was 10 at. %. Before deposition, a pre-sputtering clean was performed on both targets for 10 min to remove any contamination on the targets. During the deposition of the CuNd film, the base pressure in the deposition chamber was $5 \times 10^{-6}$ torr and the working pressure was $4 \times 10^{-3}$ torr with a fixed 20 sccm Ar flow. The used DC plasma power for the Cu target was 100 W and RF power for the CuNd target was 30 W. The distance between the targets and the substrate was 10 cm. The ambient temperature was used as the substrate temperature. The concentration of Nd was determined to be ~2.2 at. % using a high-resolution X-ray photoelectron spectroscopy (XPS; ULVAC PHI 5000 VersaProbe, Inc., Chigasaki, Japanese). In addition, pure Cu films were prepared under the same conditions for comparison. The deposition thickness of both Cu and CuNd films was approximately 100 nm, determined using an α-step 200 profilometer. The formation area of the metal electrode in the MIS capacitor was $9.0 \times 10^{-4}$ cm$^2$.

After deposition, a part of the sample was annealed in a vacuum with a pressure of $1 \times 10^{-4}$ Pa at a temperature of 425 °C for 1 h. The four-point probe (FPP) method was used to measure the sheet resistance of the films after annealing. The surface morphology of the samples was observed by a JEOL JSM-7000 field-emission scanning electron microscope (SEM) at an accelerating voltage of 10 kV. The cross-sectional structures of the CuNd/SiOCH sample before and after annealing were analyzed using a transmission electron microscopy (TEM; 300 kV, Hitachi HD-2300A, Hitachi–Science &Technology, Tokyo, Japanese).

The as-deposited and annealed MIS capacitors with Cu-2.2 at. % Nd/porous low-*k* SiOCH/*p*-Si structure were used to measure electrical characteristics and reliability. Capacitance-voltage (*C–V*) characteristics were measured using a precision impedance meter of model Agilent 4284A, and current-voltage (*I–V*) and time-dependent dielectric breakdown (TDDB) tests were measured using an electrometer (Keithley, 6517A, Austin, TX, USA). A nitrogen gas purge was carried out during the measurement to avoid moisture absorption and metal gate oxidation.

## 3. Results and Discussion

Figure 1 presents the resistivity of the Cu and CuNd films before and after annealing (425 °C/1 h). The resistivity of the CuNd film was measured as 9.6 µΩ cm, which was higher than that of the as-deposited Cu samples (7.2 µΩ cm). With the addition of Nd in the Cu film, the Cu grain was refined, increasing resistivity. According to our study, the resistivity of the CuNd film increased with increasing concentration of Nd. The increase rate was estimated to be 2.65 µΩ cm/at. % Nd. After annealing at 425 °C, the resistivity of both Cu and CuNd films decreased. The resistivity of CuNd films was measured as 3.7 µΩ cm, which is still higher than that of pure Cu films (2.9 µΩ cm). However, the decreasing rate was larger for CuNd films (~61.5 %) compared to Cu films. After annealing, the defects were annihilated and grain increased in the Cu films, leading to the decrease in sheet resistance. For CuNd films, the above two phenomena induced by annealing were pronounced due to the segregation of alloy content (Nd), resulting in a dramatic decrease in sheet resistance.

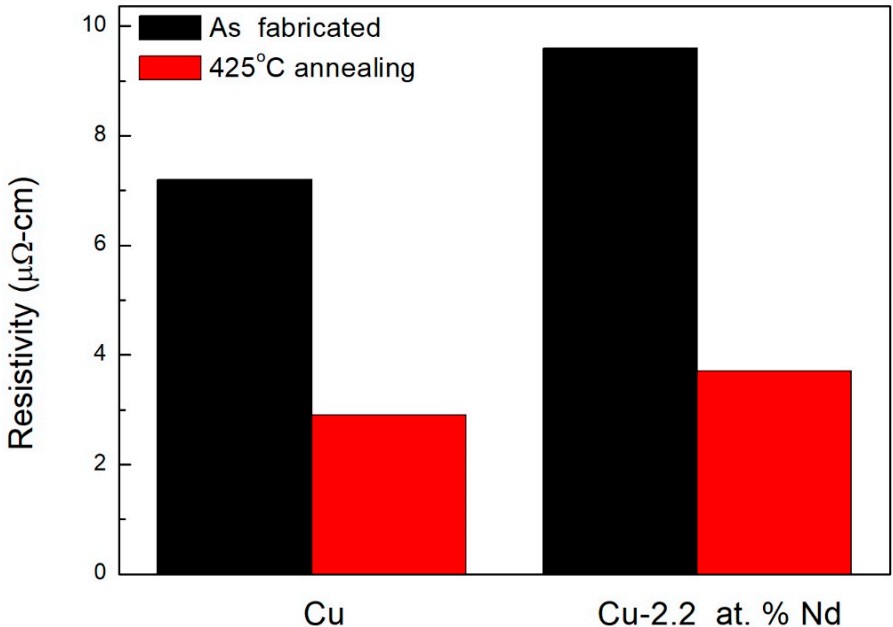

**Figure 1.** Sheet resistances of Cu and Cu-2.2 at. % Nd alloy films deposited on porous low-*k* SiOCH films before and after annealing at 425 °C.

Figure 2a,b displays the SEM top-down view of Cu and CuNd films on the porous low-*k* SiOCH film, respectively, before and after annealing at 425 °C. For both Cu and CuNd samples before annealing, no significant difference in the surface morphology was detected. However, after annealing, the surface was rough and had many pin holes of pure Cu film directly deposited onto the porous low-*k* SiOCH film. After annealing at a high temperature, Cu atoms migrated along the boundaries of Cu grains to form pin holes at the interfaces of Cu grains. This also resulted in Cu agglomeration and Cu diffusion into the porous low-*k* SiOCH film. In contrast, the annealed CuNd sample deposited onto the porous low-*k* SiOCH film still had a smooth and pin-hole-free surface. The observed grain size of the CuNd sample was smaller than that of the pure Cu sample, suggesting that Cu mass transfer and grain growth during annealing are inhibited for the CuNd sample. This is likely caused by the segregation of Nd atoms at Cu grain boundaries under thermal stress [12]. This suggests that the thermal diffusion rate of Nd is larger than that of Cu.

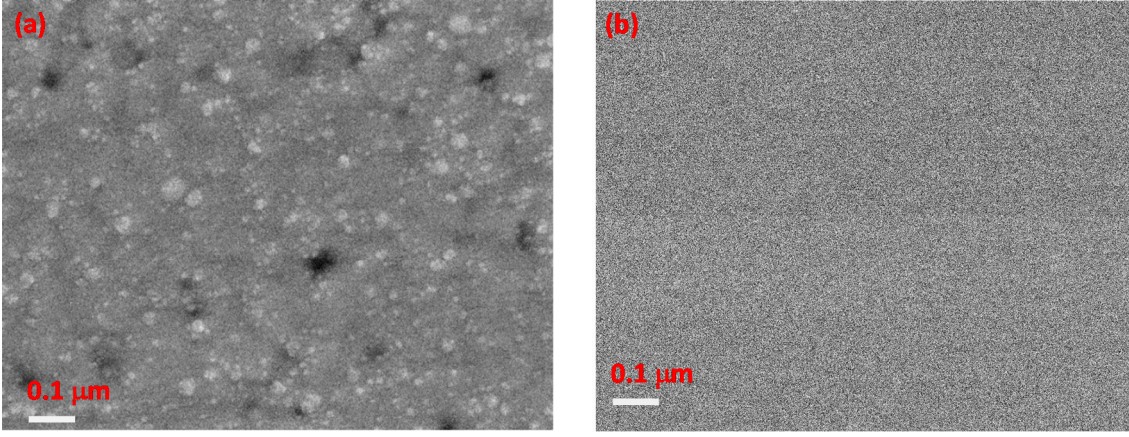

**Figure 2.** SEM top-down images of 425 °C annealed Cu/porous low-*k* SiOCH structure (**a**) without and (**b**) with Nd addition to Cu.

Figure 3 plots the *C–V* curves of the MIS capacitors with the CuNd metal gate and the porous low-*k* SiOCH film before and after annealing at 425 °C. The measured capacitances were accumulation, transition, and depletion capacitances as voltage swept from negative to positive. After annealing, the measured accumulation capacitance decreased. Additionally, a stretched *C–V* curve was observed for the as-fabricated MIS capacitor. Annealing improved the stretched shape of the *C–V* curves. These features indicate that the as-fabricated MIS capacitor with the CuNd metal gate and the porous low-*k* SiOCH film possessed charges at the interface. Using an annealing process, these interface charges were partially passivated. From the measured *C–V* curves, flat-band voltage ($V_{fb}$) was determined. $V_{fb}$ shifts represent charges introduced to the samples. The shift magnitude and direction depend on the amount and polarity of charges, respectively [18]. In the CuNd-gate MIS capacitors before and after annealing, their $V_{fb}$ values were similar; therefore, the $V_{fb}$ shift was negligible. By contrast, in the Cu-gate MIS capacitors annealed under the same condition, a pronounced $V_{fb}$ shift (~12.3 V) was observed. Additionally, its shift tended toward negative voltage, showing that positively charged defects were introduced into the test MIS capacitor during an annealing. These introduced positively-charged defects in the annealed Cu-gate MIS capacitor have been widely reported to be Cu ions [19]. A barrier layer, therefore, is required between the Cu and the porous low-*k* SiOCH film. A negligible $V_{fb}$ shift was detected as the CuNd/porous low-*k* SiOCH integration underwent annealing, suggesting that the diffusion of Cu ions due to thermal stress was blocked. This blocking capacity is thought to result from a diffusion barrier at CuNd/SiOCH interface. We therefore speculate that the diffusivity of Nd is higher than that of Cu upon annealing at a high temperature of 425 °C. Before Cu ions diffuse to the surface of a porous low-*k* SiOCH film, Nd atoms with a faster diffusion rate react with SiOCH film to form a thin barrier layer, which is thermally stable and can limit further diffusion of Cu ions into the porous low-*k* SiOCH film.

The leakage current was monitored by sweeping the voltage in a ramp voltage stress (RVS) measurement. Negative voltages were applied on the metal gate of MIS capacitors. The applied voltage was transformed to the electric-field (*E*) by dividing by the thickness of the porous low-*k* SiOCH film. Figure 4 plots the leakage current density versus the applied field (*I–E*) curves of porous low-*k* SiOCH films with the CuNd metal gate before and after annealing. Similar to our previous work [18,19], the *I–E* curves of porous low-*k* SiOCH films with CuNd metal gate exhibited three stages. In the first stage, the leakage current increased with the applied field. Next, the increase in the leakage current slowed down as the applied field increased. In the third stage, the measured leakage current suddenly jumped by at least three orders of magnitude. At that time, the applied field was defined as the breakdown field of a tested porous low-*k* SiOCH film. As compared in Figure 4, annealing the MIS capacitor with the CuNd metal gate and the porous low-*k* SiOCH film decreased the leakage current before breakdown and enhanced the breakdown field. Additionally, the transition fields from the first stage to the second

stage in the *I–E* curves were similar for both as-fabricated and annealed MIS samples. For a Cu-gate MIS capacitor annealed at 425 °C for 1 h, its measured *I–E* behavior was different, as the leakage current increased and the breakdown field decreased. These degradations were reportedly caused by the diffusion of Cu into the porous low-*k* SiOCH film by a thermal annealing [20]. A decreasing leakage current for the annealed MIS capacitor with CuNd metal gate and the porous low-*k* SiOCH film suggests that the thermal diffusion of Cu is suppressed. Annealing modifies and strengthens the interface between CuNd and the porous low-*k* SiOCH film, thereby decreasing leakage current and enhancing the breakdown field.

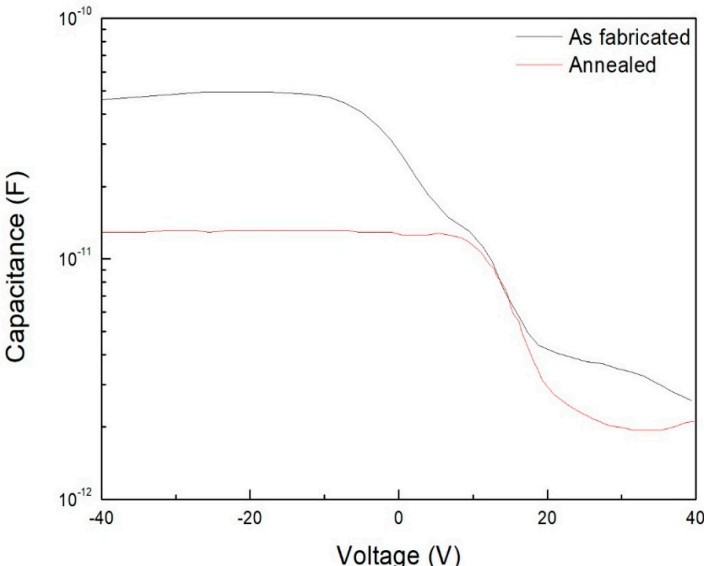

**Figure 3.** *C–V* curves of Cu-2.2 at. % Nd alloy films deposited on porous low-*k* SiOCH films before and after annealing at 425 °C.

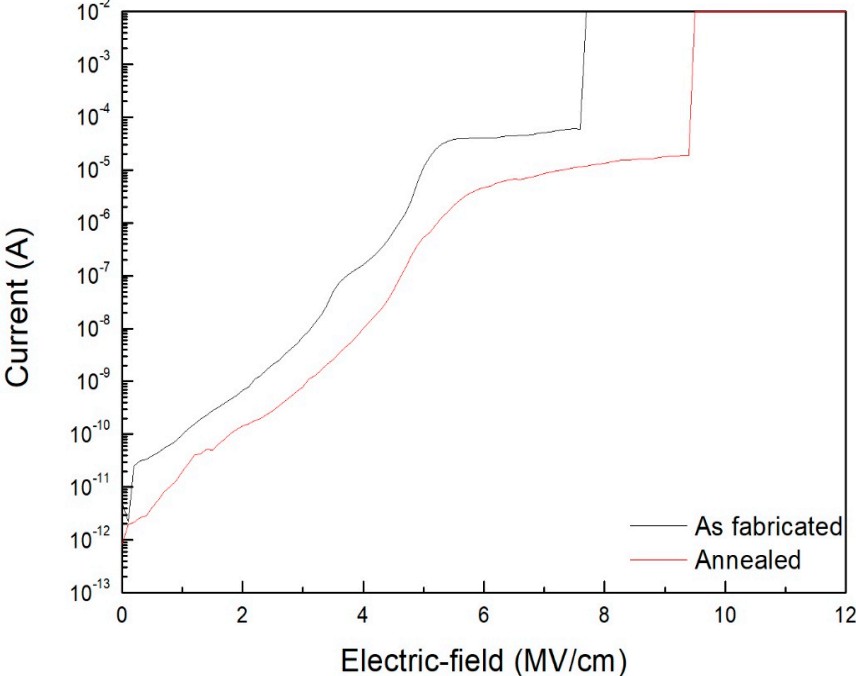

**Figure 4.** *I–E* plots of Cu-2.2% Nd alloy films deposited on porous low-*k* SiOCH films before and after annealing at 425 °C.

The dielectric breakdown fields of the porous low-*k* SiOCH film in both Cu-gate and CuNd-gate MIS capacitors were measured from 10 samples. Figure 5 compares the changes in the breakdown field for the Cu-gate and CuNd-gate MIS capacitors with the same porous low-*k* SiOCH films before and after annealing. Annealing resulted in different trends in the Cu-gate and CuNd-gate MIS capacitors. The breakdown field significantly decreased for the annealed Cu-gate MIS capacitor, whereas it improved to 9.25 ± 0.40 from 8.05 ± 0.65 MV/cm for the annealed CuNd-gate sample. The thermal diffusion Cu atoms/ions into the porous low-*k* SiCOH film upon annealing was responsible for a decreasing breakdown field for the Cu-gate MIS capacitor [20]. An improved breakdown field in the annealed CuNd-gate MIS capacitors suggests that the thermal migration of Cu atoms/ions is inhibited upon annealing. The interface between CuNd and the porous low-*k* SiOCH film is modified by thermal annealing, providing an additional resistance to failure. The possible mechanism is caused by the formation of the interfacial oxide due to the reaction between Nd atoms and SiCOH film.

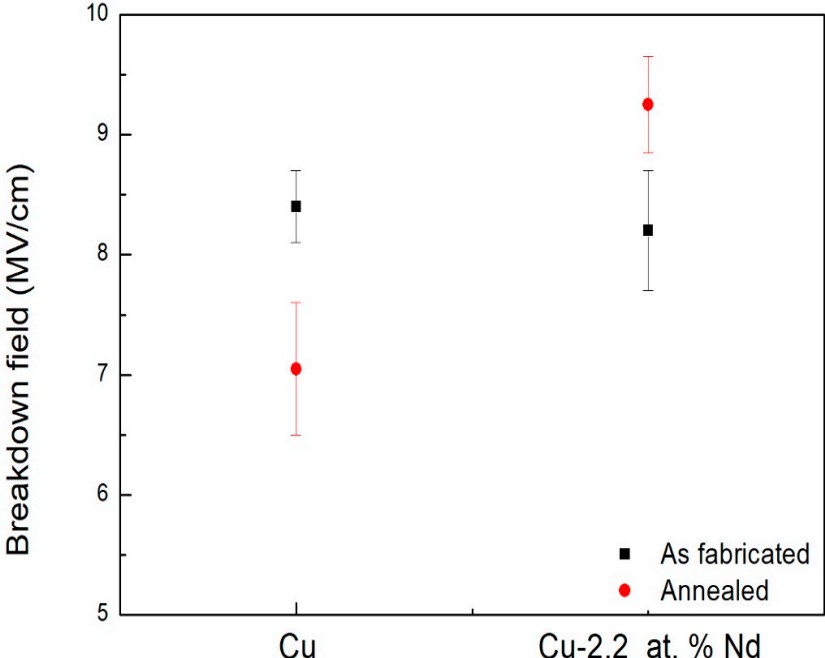

**Figure 5.** Breakdown fields of porous low-*k* SiOCH films in Cu-gate and CuNd-gate MIS capacitor before and after annealing.

The breakdown times (TTFs) of the CuNd-gate MIS capacitors with the porous low-*k* SiCOH film were measured using a TDDB test. Twelve samples were measured in each condition. Figure 6 plots the median breakdown times of the CuNd-gate MIS capacitors before and after annealing as a function of the stressing electric field. The results obtained from both as-fabricated and annealed CuNd-gate MIS capacitors indicated that stronger applied electric fields were associated with lower dielectric breakdown times. The applied electric field, therefore, is an important role dominating dielectric breakdown. Comparison of the TDDB results of the as-fabricated and annealed CuNd-gate MIS capacitors indicated that annealing increased the breakdown times, similar to the results of the breakdown field. Additionally, the lines of the breakdown time versus electric field shown in Figure 6 are almost parallel for as-fabricated and annealed CuNd-gate samples, suggesting that the electric field acceleration factors are not significantly impacted by annealing. The electric field acceleration factors ($\gamma$) for the E model (TTF = A exp ($-\gamma$ E)) were determined to be 2.21 ± 0.18 and 2.02 ± 0.24 cm/MV for the as-fabricated and annealed CuNd-gate MIS capacitors, respectively. As Cu-gate MIS capacitors were annealed under the same conditions, different features in the TDDB results were observed [21]. For the annealed Cu-gate MIS capacitors, the TDDB breakdown time and electric field acceleration factor significantly degraded, showing that the breakdown mechanism changed from intrinsic to

extrinsic modes. The extrinsic degradation was attributed to the Cu injection mechanism [22]. For the annealed CuNd-gate MIS capacitors, the degraded TDDB characteristics were not detected, suggesting that the Cu injection mechanism is not involved in dielectric breakdown. An improved breakdown time implies that annealing a CuNd/porous low-*k* SiOCH film stacked sample strengthened its dielectric strength. The formation of an interfacial oxide layer is the most likely mechanism. This formation layer can limit the migration of Cu ions and promotes adhesion, thus slowing down TDDB breakdown.

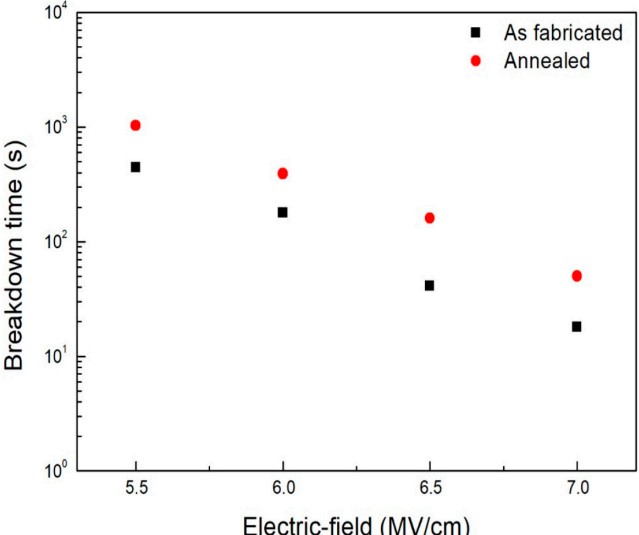

**Figure 6.** Dielectric breakdown times of porous low-*k* SiOCH films with CuNd before and after annealing as a function of applied electric field.

Figure 7 shows the cross-sectional TEM image of 425 °C annealed CuNd films deposited directly onto the porous low-*k* SiOCH film. After annealing at 425 °C, an ultrathin white-colored layer appeared at the interface between CuNd and the porous low-*k* SiOCH film. In the as-deposited sample that did not experience heat treatment or annealed Cu-gate sample, no such layer was seen. This formation layer can be inferred as the reaction between Nd atoms and SiOCH film under a thermal annealing. We observed that this self-forming layer was not uniform and the thickness was estimated to be approximately 8 to 12 nm.

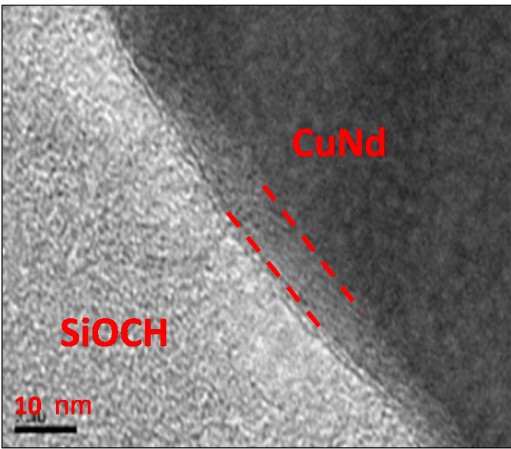

**Figure 7.** Cross-sectional TEM image of annealed (at 425 °C for 1 h) Cu-2.2% Nd film directly deposited onto a porous low-*k* SiOCH film. The region between the dashed lines is newly-formed interfacial layer.

## 4. Conclusions

In this study, Cu-2.2 at. % Nd alloy films were deposited by physical vapor deposition onto the porous low-*k* SiOCH film and then annealed at 425 °C. By annealing, a self-forming barrier layer formed at the interface between CuNd alloy and the porous low-*k* SiOCH film. As a result, the leakage current, breakdown field, and TDDB reliability of the porous low-*k* SiOCH film improved. Consequently, adding Nd alloying element in the Cu film to form a self-forming barrier layer at the surface of the porous low-*k* SiOCH film is a promising process for advanced Cu interconnects.

**Author Contributions:** Conceptualization, Y.-L.C.; Data curation, J.-S.F.; Investigation, W.-F.P.; Methodology, Y.-L.C. and C.-Y.L.; Project administration, Y.-L.C.; Validation, J.-S.F.; Visualization, J.-S.F.; Writing – original draft, Y.-L.C.; Writing – review & editing, G.-S.C. All authors have read and agreed to the published version of the manuscript.

**Funding:** The authors would like to thank the Ministry of Science and Technology of the Republic of China, Taiwan, for financially supporting this research under contract MOST 107-2221-E-260-001-MY2.

**Conflicts of Interest:** The authors declare no conflict of interest.

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
