# Peer review of "Electrical and Reliability Characteristics of Self-Forming Barrier for CuNd/SiOCH Films in Cu Interconnects"

_coatings, doi:10.3390/coatings10020155_

Round 1

Reviewer 1 Report

The authors have shown an interesting and original work. I would like to mention that the manuscript needs further improvement in the points mentioned below:

The purity ratios of Cu and Nd targets are not mentioned in the "Experiments" section. It would be useful to indicate these values. It would be supportive for the manuscript in case the authors could provide information about the phases obtained in the annealed sample. Phase identification by TEM would be helpful in improving the article. Manipulating the annealing parameters by changing the duration or temperature could help in forming an approach for the trend that the self-forming barrier would exhibit. Can the authors provide information about results from annealing treatments carried out for different durations or different temperatures?

Author Response

The purity ratios of Cu and Nd targets had been added in the "Experiments" section in the revised manuscript. Phase identification by TEM is on-going, we will provide this information in the following work. The annealing conditions were provided in the "Experiments" section. We only studied the effect of the annealing temperture (300~800C), no carried out for different durations. Thermal budget (Temp & Duration) effect on self-forming barrier is still researching.

Reviewer 2 Report

The authors presented the electrical and reliability studies on CuNd/SiOCH films in Cu Interconnects. The research is interesting, but I would request the authors to address the following points before I could recommend this manuscript for publication:

1. Was there any surface morphology study performed on the films, such as AFM? 

2. If the authors could compare pure Cu with CuNd alloy films side by side for characterization results, such as in figures 3-4, that would be beneficial for the readers. It appears from the text that the authors may have performed this analysis as well.

3. Did the authors study other than 2.2- atomic % of Nd in the alloy? Is there any insight on how varying the % of Nd would affect the performance, so that it can be taken into design consideration?

4. In the breakdown studies, was the increased resistance of CuNd alloy taken into consideration? If not, was it insignificant?

5. Please add TEM images for Cu samples and also pre-anneal CuNd samples  in Fig 7 so that the improvement after annealing can be understood well. Also, figure caption should mention the region between the dashed lines between CuNd and SiOCH.

6. The SiOCH film was estimated to be of ~130nm thickness, and as per Fig 7, the added interfacial oxide film after annealing the CuNd-deposited sample had a thickness of ~10 nm (8-12 nm per text). If that increase in the thickness of the effective oxide barrier is taken into consideration, how does the C-V/breakdown characteristics look like? i.e, is that improvement coming mostly from the barrier thickening, or the blocking of Cu migration also a major factor here yet? If the authors can determine the nature (i.e. elemental composition) of the oxide and estimate its dielectric constant, they could approximate the C-V performance of the stack within an order of magnitude value.

Author Response

1. No, we only checked SEM, no AFM.

2. For as-deposited film, no difference in SEM. Therefore, we only presented the annealed results in Fig.2.

3. For Various % of Nd, we only check their resistance ( Four-point probe (FPP) ). The increase rate was estimated to be 2.65 Ω-cm/atomic % Nd . This information was provided in the text. Its effects on the electrical characteristics and reliability will be further studied in the future research.    

4. Based on our previous study, the breakdown was less impacted by the increased resistance of CuNd alloy.  

5. TEM images  in Fig 7 had been revised. Figure caption had been added  the region between the dashed lines .

6. We think that the improvement for CuNd alloy film is from the barrier thickening and the blocking of Cu migration. Now, we canot identify their contribution.  This issue will be further studied.